# Koopman Operator Based Dynamical Similarity Analysis for Data-driven Quantification of Distance between Dynamics

**Shunsuke Kamiya, & Masafumi Oizumi** [*]
Graduate School of Arts and Sciences
The University of Tokyo
Tokyo, Japan
{mathmrk,c-oizumi}@g.ecc.u-tokyo.ac.jp

**Jun Kitazono**
Department of Data Science
Yokohama City University
Kanagawa, Japan
c-kitazono@g.ecc.u-tokyo.ac.jp

## Abstract

Quantifying distance between two dynamical neural systems is a fundamental problem in neuroscience and machine learning fields. Neural dynamics are known to possess nonlinear features, which makes comparison between systems difficult. Recently, a promising method to quantify distance between dynamics called Dynamic Similarity Analysis (DSA) is proposed (Ostrow et al., 2023), which measures distance between matrices approximating linear operators defined in time delay embedded space and thus takes nonlinearity into accounts. Although being a strong method, DSA is not free from problems, including obscure interpretability, failure to satisfy the triangle inequality among matrices of different dimensions, and long computational time. To address these problems, we propose a modified novel version of DSA. Our proposed DSA measures distance between approximated Koopman operators, which has better interpretability as a linear operator that drives dynamics in a mapped space. The distance measure adopted in our method satisfies the triangle inequality even between matrices of different dimensions. This distance measure also allows extremely fast computational time. We applied our method and DSA to Lorenz system (Lorenz, 1963) of various parameters, and found that our method revealed clusters with respect to parameters and dynamical properties, while DSA failed to do so. With theoretical underpinnings of Koopman operators and matrix distance, we propose our method as an effective method to quantify distance between dynamics.

## 1 Introduction

Comparing representations of two or more neural systems has become an important problem in neuroscience and machine learning fields. For instance, comparison between brain activity of different subjects (O'Connell & Chun, 2018; Hebart et al., 2020), or between brain activity and deep neural network outputs (Yamins et al., 2014; Cichy et al., 2016; Horikawa & Kamitani, 2017), may provide better understandings of common mechanism of neural information processing, or revealing unknown neural working principles. This information processing is essentially reduced as the activity of neurons —they transmit information to each other and change their activities dynamically. As information processing appears as a time series of the dynamical system, capturing dynamical features of neural activity is vital in comparing foundational computations that systems perform. Comparing dynamical systems, or more specifically, quantifying distance between dynamical systems is thus an important question, possibly opening the door to comparing many different neural systems and information processing machines such as deep neural networks (Sexton & Love, 2022; Fu et al., 2023) or large language models (Kawakita et al., 2023; Wang et al., 2023).

One of the important aspects to consider dynamics of information processing systems is nonlinearity. It has been known that neural systems and machine learning models own nonlinear representational features (Rabinovich & Muezzinoglu, 2010) and that they play important roles in performing com-

---

[*] Use footnote for providing further information about author (webpage, alternative address)

putations (Breakspear, 2017; Freyer et al., 2012). Thus, when comparing two information processing systems, methods that take nonlinearity into account are favorable rather than utilizing techniques to compare linear dynamical systems, which have a rather long research history in the field of system identification (Hanzon & Marcus, 1982; Afsari & Vidal, 2013). A possible approach for comparison taking account of nonlinearity is the use of the Koopman operator method (Koopman, 1931; Mezić, 2005; Brunton et al., 2022), which has become one of the major data-driven strategies to tackle nonlinear dynamics.

Recently, one such method, called Dynamic Similarity Analysis (DSA), has been invented (Ostrow et al., 2023). DSA is a data-driven method that measures the distance between a pair of time series data. Briefly, this method takes two steps: the first step is Hankel Alternative View of Koopman (HAVOK) analysis (Brunton et al., 2017), where one finds a matrix for each dynamics as an approximation of a linear operator in time delay coordinate space. The next step is the measuring step, and one quantifies a distance between the obtained pair of matrices. This quantification makes use of what is called Procrustes analysis of vector fields (PAVF), where a change of basis of coordinate is regarded as invariant. The authors claim that DSA succeeded in distinguishing dynamics that are similar in a geometric sense, and identifying dynamical systems of similar dynamical properties that possess different geometrical features (Ostrow et al., 2023), both of which are unsuccessful with existing methodologies that utilize geometrical characteristics of state space, such as orthogonal Procrustes analysis.

Although DSA is a promising and distinctive method to compare dynamics as mentioned above, it is not free from problems. Firstly, its interpretability is not clear. This lack of interpretability of DSA can be attributed to that of HAVOK, whose theoretical foundations and backgrounds have been studied (Arbabi & Mezić, 2017; Hirsh et al., 2021; Kamb et al., 2020) but are yet to be fully explained. Secondly, the distance measure, PAVF, cannot satisfy the triangle inequality when applied to quantify matrices of different dimensions. This may be problematic, as we often apply based on the dissimilarity various machine learning techniques that are based on metric properties of the dissimilarity measure (Williams et al., 2021). Thirdly, to find distance using PAVF, one has to use optimization algorithms including neural networks (Ostrow et al., 2023) or iterative optimization techniques over Stiefel manifolds (Edelman et al., 1998; Sato & Iwai, 2013). This can cause a long computational time, limiting the applicability of the method.

To address these problems, here we propose an altered version of DSA to quantify distance between two dynamics (Fig. 1). We make two alterations. Firstly, our method compares two matrices that are direct approximations of the Koopman operator (Koopman, 1931; Mezić, 2005; Bevanda et al., 2021; Brunton et al., 2022). Having a clear meaning as linear operators that dominate the dynamics in nonlinearly transformed spaces, the Koopman operator possesses better interpretability and clarity in quantifying distance between dynamics. We can also employ kernel methods to enable flexible representation for the dynamics and at the same time simpler calculations (Baddoo et al., 2022).

Secondly, we modified the distance metric originally proposed in DSA and defined a distance measure called Modified Procrustes analysis of vector fields (M-PAVF). This metric respects the triangle inequality even between matrices of different sizes and successfully evades the problem in the original metric. This modified metric is also advantageous in its computational simplicity, as one can find the metric massively faster than the metric in DSA, preserving similar tendency as the DSA metric.

As an application, we employed our version of DSA and the original DSA to time series data of Lorenz system (Lorenz, 1963) of different parameters to see how these methods quantify distances between dynamics. The results show that our method showed clusters of dynamical data of the same parameter, while the original DSA could not. What is more, only did our method succeed in detecting the closeness with respect to dynamics' dynamical properties of different parameters, revealing that our version of DSA can be a favorable measure to quantify distance between dynamics.

## 2 BACKGROUNDS

Before describing our method, we first explain some backgrounds about data-driven approaches of nonlinear dynamical analysis. Our study makes use of concepts of the Koopman operator and related methodologies including Extended Dynamic Mode Decomposition (EDMD), kernel EDMD

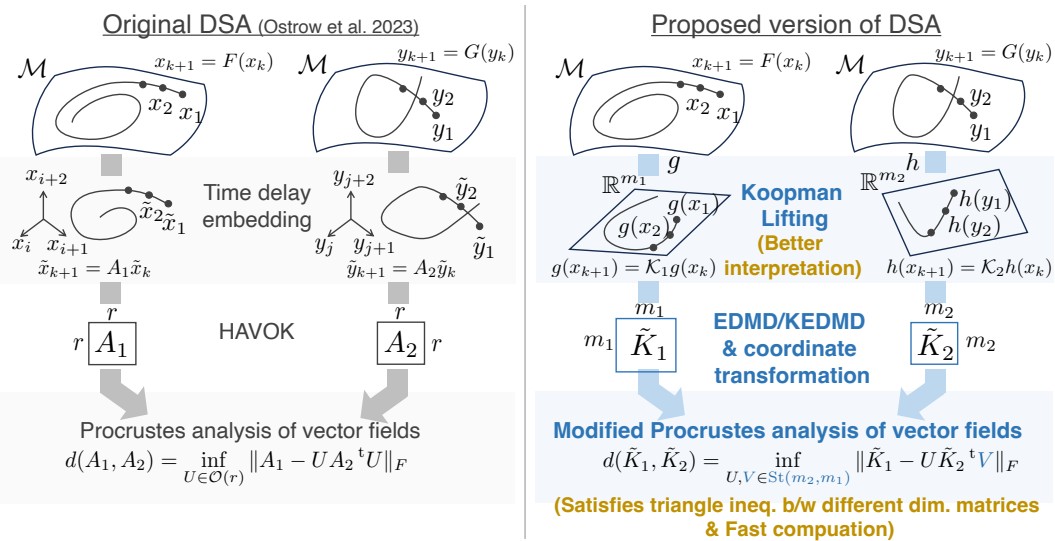

Figure 1: **Comparison of overview of original DSA and our version of DSA.**

(KEDMD), and Hankel Altered View of Koopman analysis (HAVOK). In what follows, we are to explain each of these and the original DSA briefly.

## 2.1 KOOPMAN OPERATOR

Initially proposed a century ago in Koopman (1931) and after rediscovery in Mezić (2005), the Koopman operator has become one of the mainstream approaches in the analysis of nonlinear dynamical systems (Mezic, 2013; Bevanda et al., 2021; Brunton et al., 2022). In a nutshell, the Koopman operator is a linear operator that drives nonlinearly mapped dynamics. In more detail, assume a discrete-time dynamical system $\{z_n\}_{n=1}^{\infty}$ on a manifold $\mathcal{M} \subset \mathbb{R}^d$ ($d \in \mathbb{N}$) is defined as,

$$z_{n+1} = F(z_n), \quad n = 1, 2, \cdots, \tag{1}$$

with $F : \mathcal{M} \to \mathcal{M}$. Here $F$ may be a nonlinear transform, and thus the dynamics may be difficult to analyze any further. To get a better perspective, consider a dynamics that is given as a transformation of $z_n$ through a function $g : \mathcal{M} \to \mathbb{R}$,

$$g(z_{n+1}) = g \circ F(z_n), \quad n = 1, 2, \cdots. \tag{2}$$

The composition of the functions on the right-hand side can be regarded as an operator $\mathcal{K} := g \circ F$. This $\mathcal{K}$ is called the **Koopman operator**. When we set $g$ as a member of a Hilbert space $\mathcal{F}$ with an inner product $\langle \cdot, \cdot \rangle$, and assume $g \circ F \in \mathcal{F}$ for any $g \in \mathcal{F}$, we readily see that the Koopman operator is a linear operator on $\mathcal{F}$.

The merit of the Koopman operator is that it transforms a *nonlinear* dynamics defined as Eq. (1) into a *linear* dynamics. For $g \in \mathcal{F}$ and $\{g(z_n)\}_{n=1,\cdots}^{\infty}$,

$$g(z_{n+1}) = \mathcal{K}g(z_n), \tag{3}$$

which shows that $\mathcal{K}$ has a clear mathematical meaning as a linear operator that drives the mapped dynamics $\{g(z_n)\}$. This mapping $g$ is often called an **observable** (Williams et al., 2015).

Now with a linear equation Eq. (3) at hand, one may expect to examine properties of the dynamical system in a simple way. For example, one may want to find eigenvalues and eigenfunctions of $\mathcal{K}$ to gain an insight into the dynamics, or make a prediction of $z_n$ at some unseen time point. Unfortunately, it is usually difficult to do so using Eq. (3), as $\mathcal{K}$ is a linear operator in an (generally) infinite dimensional space $\mathcal{F}$. Instead, one has to think of a finite dimensional approximation of $\mathcal{K}$, i.e., one needs to limit the functional space to a finite dimensional subspace. This technique is called Extended Dynamic Mode Decomposition (EDMD) (Williams et al., 2015) and is discussed next.

## 2.2 EXTENDED DYNAMIC MODE DECOMPOSITION (EDMD)

EDMD (Williams et al., 2015) is a data-driven technique to approximate the action of the Koopman operator. As mentioned above, it is often difficult to find the eigenvalues and eigenfunctions of the linear operator $\mathcal{K}$, and one needs to confine the action of the operator in a finite dimensional space. That is, we take an $N$ dimensional space $\mathcal{F}_N$ ($N \in \mathbb{N}$) and consider approximating $\mathcal{K}$ with a linear operator $\mathcal{K}' : \mathcal{F}_N \to \mathcal{F}_N$. In EDMD, we take $\mathcal{F}_N$ as a span: $\mathcal{F}_N := \mathrm{span}(\psi_1, \cdots, \psi_N)$, where the functions $\psi_1, \cdots, \psi_N \in \mathcal{F}$ are arbitrary chosen. Each of the functions $\psi_i$, $(i = 1, \cdots, N)$ is called a *basis function*, and the combination of $N$ functions is called a *dictionary*. Some typical choices of such a dictionary can be monomial functions, or orthogonal polynomial functions such as Hermite polynomials (under $\mathcal{F}$ being an $L^2$ space with respect to some measure on $\mathcal{M}$).

To approximate $\mathcal{K}$ with $\mathcal{K}' : \mathcal{F}_N \to \mathcal{F}_N$, we take a function $h \in \mathcal{F}_N$, and think of the image $\mathcal{K}'h$ as the projection of $\mathcal{K}h$ onto $\mathcal{F}_N$. This projection is necessary as $\mathcal{K}h$ is not always on $\mathcal{F}_N$ and one has to approximate $\mathcal{K}h$ with a point on $\mathcal{F}_N$. Some calculation will show that if $\{\psi_i\}_{i=1}^N$ is linearly independent and $h$ is expressed as $h =: \sum_{i=1}^N c_i \psi_i$ with $c_1, \cdots, c_N \in \mathbb{R}$, then $\mathcal{K}'h$ is written as $\sum_{i=1}^N c_i' \psi_i$, with

$$^{\mathrm{t}}(c_1', \cdots, c_N') = K \,^{\mathrm{t}}(c_1, \cdots, c_N). \tag{4}$$

Here, $K$ is a matrix defined as

$$K = \,^{\mathrm{t}}(AG^{-1}), \tag{5}$$

where

$$(A)_{ij} = \langle \mathcal{K}\psi_i, \psi_j \rangle, \quad (G)_{ij} = \langle \psi_i, \psi_j \rangle. \tag{6}$$

Now we obtained the approximated form of $\mathcal{K}$ on the finite dimensional subspace $\mathcal{F}_N$, which can be conveniently described using linear algebra.

How can we find $A$ and $G$ using dynamical data? With the time series data $\{x_n\}_{n=1}^M \subset \mathcal{M}$ ($M \in \mathbb{N}$), we can approximate $A$ and $G$ utilizing $\mathcal{K}\psi_i(x_n) = \psi_i(x_{n+1})$ ($n = 1, \cdots, M-1$) and the Ergodic hypothesis. This approximation, however, can be problematic, with a large number of the basis functions ($N$) causing enormous memory load and difficulty in taking the matrix inverse ($G^{-1}$). We often face this large $N$ problem particularly when the dynamics is high-dimensional, since the number of the basis functions grows immensely due to combinatorial explosion. Kernel Extended Dynamic Mode Decomposition (KEDMD) (Williams et al., 2016; Klus et al., 2020) is a method proposed to alleviate such a problem, where kernel methods come in and drastically facilitate the matrix computations. We next look at this method.

## 2.3 KERNEL EXTENDED DYNAMIC MODE DECOMPOSITION (KEDMD)

Basic concepts of KEDMD are almost identical to those of EDMD, except that $\mathcal{F}$ is taken to be a reproducing kernel Hilbert space (RKHS) $\mathcal{H}$ with respect to a positive definite kernel $k : \mathcal{M} \times \mathcal{M} \to \mathbb{R}$ and the inner product $\langle \cdot, \cdot \rangle$ defined using $k$. In short, $k$ is a generalization of inner products, or similarity measure, of a pair of points in $\mathcal{M}$, and the RKHS is a Hilbert space of functions where functions of the form of $k(\cdot, x)$ ($\forall x \in \mathcal{M}$) and their linear sums reside and for any $f \in \mathcal{H}$, $\langle f, k(\cdot, x) \rangle = f(x)$ holds. In KEDMD, each $\psi_i$ is set to be $\psi_i(\cdot) = k(\cdot, x_i)$, $i = 1, \cdots, M$ (Klus et al., 2020; Baddoo et al., 2022; Rosenfeld et al., 2022). Note that the subscripts go over 1 to $M$, not 1 to $N$. Using the properties of the kernel function, $A$ and $G$ in Eq. (6) is written as

$$(A)_{ij} = k(x_{i+1}, x_j), \quad (G)_{ij} = k(x_i, x_j). \tag{7}$$

There are two merits in applying kernel methods to EDMD. First, kernel methods allow us to deal with nonlinear regressions in a tractable manner, thanks to the *kernel trick*, enabling flexible representation of systems (Shawe-Taylor & Cristianini, 2004). Second, computational loads lessen because one only has to take $M$ (the number of time points) basis functions at maximum. This drastically improves computational complexity if $N$ is impermissively large, especially due to the matrix inversion process $G^{-1}$ taking $O(M^3)$ steps instead of $O(N^3)$. It can still be hard with $M$ basis functions in computing $K$ as in Eq. (5) if $M$ is not small enough. But there exist many techniques to evade this problem utilizing essentially row rank property of Gram matrices $G$, such as Nystrom methods (Drineas & Mahoney, 2005; Sun et al., 2015; Zhang & Kwok, 2010). Recently, in KEDMD literature, a type of Nystrom method was proposed as a part of an algorithm named linear and nonlinear disambiguation optimization (LANDO) (Baddoo et al., 2022). We adopt this technique in our computation of $K$.

## 2.4 Hankel Alternative View of Koopman (HAVOK) Analysis

HAVOK analysis was originally proposed as a method to decompose a chaotic system into a linear system and intermittent forcing (Brunton et al., 2017). HAVOK makes use of a classical technique called time delay embedding (Kim et al., 1999; Pan & Duraisamy, 2020), where one stacks time series data $\{x_n\}_{n=1}^{M} \subset \mathbb{R}^d$ to obtain what is called a Hankel matrix:

$$H_{1:M-h+1}^{(h)} := \begin{pmatrix} x_1 & x_2 & \cdots & x_{M-h+1} \\ x_2 & x_3 & \cdots & x_{M-h+2} \\ \vdots & & & \vdots \\ x_h & x_{h+1} & \cdots & x_M \end{pmatrix}, \tag{8}$$

where $h \in \{1, \cdots, M\}$. In HAVOK, one tries to extract a linear relationship between consequent two time steps on this time delay embedded coordinate, but does so using what can be thought as the "basis" for this time series. To compare between two time steps, $H_{1:M-h}^{(h)}$ and $H_{2:M-h+1}^{(h)}$ are taken, which we hereafter denote by $H_1$ and $H_2$, respectively, for notational simplicity. One applies singular value decomposition (SVD) of rank $r$ ($r \leqslant \min(hd, M-h)$) to these matrices: $H_1 \simeq U_1 \Sigma_1 {}^\mathsf{t}V_1$ and $H_2 \simeq U_2 \Sigma_2 {}^\mathsf{t}V_2$. These $V_1$ and $V_2$ represent dominant structures of the time series $\{x_n\}$ and can be thought as bases for it. Then, using these bases, linear regression is performed to find a matrix $A$ such that ${}^\mathsf{t}V_2 = A\,{}^\mathsf{t}V_1$.

What does the HAVOK procedure mathematically mean? If $H_{1:M-h+1}^{(h)}$ is composed over an infinitesimally short time period (i.e., $M-h$ is small enough), $A$ is a matrix that appears in Frenet-Serret equation, which represents time variation of tangent, normal, and binormal vectors, for the dynamics in time delay embedding space (Hirsh et al., 2021). In the opposite limit of $M \to \infty$, HAVOK converges to discrete Fourier analysis when $\{x_n\}$ is periodic (Bozzo et al., 2010). For other general cases, although much work has been done for revealing mathematical properties of HAVOK (Arbabi & Mezić, 2017; Kamb et al., 2020), a clear-cut mathematical interpretation of HAVOK seems to be missing.

## 2.5 Dynamic Similarity Analysis (DSA)

DSA is a method to quantify distance between two given systems (Ostrow et al., 2023). It builds on HAVOK and a matrix distance quantification technique called Procrustes analysis of vector fields (PAVF). Given two dynamical data $\{x_n\}_{n=1}^{M_1} \subset \mathbb{R}^{d_1}$ and $\{y_n\}_{n=1}^{M_2} \subset \mathbb{R}^{d_2}$ ($M_1, M_2, d_1, d_2 \in \mathbb{N}$), the following two steps are taken:

**1. HAVOK analysis** Perform HAVOK to each of the data using the first $r \in \mathbb{N}$ rows of the Hankel matrices, and obtain matrices $A_x, A_y \in \mathbb{R}^{r \times r}$ from data $\{x_n\}_{n=1}^{M_1}$ and $\{y_n\}_{n=1}^{M_2}$, respectively.

**2. Procrustes analysis of vector fields (PAVF)** To see how close $A_x$ and $A_y$ can be as drift matrices even after a change of basis, solve the next optimization problem:

$$d(A, B) := \inf_{U \in \mathcal{O}(r)} \|A_x - U A_y\,{}^\mathsf{t}U\|_F, \tag{9}$$

to find the distance between dynamics. Here, $\mathcal{O}(r)$ is the set of orthogonal matrices of dimension $r$.

In the original paper (Ostrow et al., 2023), the authors applied DSA to simulated dynamical data and showed that DSA was able to detect the difference between dynamics that are similar in a geometric sense, and identify dynamical systems of similar dynamical properties that possess different geometrical features, both of which were impossible using existing methods.

Being a promising approach that takes nonlinearity into account and owns an empirically favorable ability to distinguish dynamical properties, DSA is not free from problems. We here list three of them. First, its interpretability is not clear. This lack of interpretability is caused by the use of HAVOK, whose theoretical foundations remain unexplained as mentioned above. Secondly, the distance measure, PAVF (Eq. (9)), does not always satisfy the triangle inequality when applied to matrices of different dimensions. PAVF was originally proposed to quantify distance between matrices of the same dimension as in Eq. (9), and under this condition satisfies the triangle inequality, which is considered favorable as a metric quantifying representational dissimilarity (Williams et al., 2021). As we are to see later, we can easily extend PAVF for matrices of different dimensions,

but that is when a problem occurs; the metric breaches the triangle inequality between matrices of different dimensions. This can be problematic, as after obtaining distance, or dissimilarity, of systems, we often apply machine learning techniques including $k$-nearest neighbors (Hastie et al., 2009) that are theoretically based on metric properties of the dissimilarity measure. Using metrics that can break the triangle inequality may cause inconsistent results in those later analyses, possibly hindering our understandings. Thirdly, to solve PAVF (Eq. (9)), one has to use optimization algorithms including neural networks (Ostrow et al., 2023) or iterative optimization techniques over Stiefel manifolds (Edelman et al., 1998; Sato & Iwai, 2013). This can cause long computational time, especially when matrix dimensions are large, and thus confines applicability. To address these problems, we propose a novel altered version of DSA, which is to be explained in detail in the next section 3.

# 3 RESULTS

## 3.1 PROPOSED MODIFICATION OF DSA

In this subsection, we explain our modified version of DSA to quantify distance between given two systems. In a nutshell, we measure distance between dynamics by calculating the distance between the approximated Koopman operators. Our method thus consists of the following two steps:

1. Apply EDMD or KEDMD to find Koopman approximant matrices for each system.

2. Perform a modified version of PAVF and compute the distance between the systems.

We are to explain each of the steps in detail in the following. See Fig. 1 for conceptual description. Let us assume that we have access to a pair of dynamical data $\{x_n\}_{n=1}^{M_1} \subset \mathcal{M}$ and $\{y_n\}_{n=1}^{M_2} \subset \mathcal{M}$, where $\mathcal{M} \subset \mathbb{R}^d$ $(M_1, M_2, d \in \mathbb{N})$. Here, the dimensions of $\{x_n\}$ and $\{y_n\}$ are set to be identical for a theoretical reason, and practically for data of different dimensions one can zero-pad dynamics as necessary.

**Apply EDMD or KEDMD** First, we perform EDMD or KEDMD to estimate the finite dimensional Koopman approximant for each system. We compute $K$ as in Eq. (5) using expressions Eq. (6) or Eq. (7) for each dynamical data $\{x_n\}_{n=1}^{M_1}$ and $\{y_n\}_{n=1}^{M_2}$, and put them $K_1$ and $K_2$, respectively.

Applying EDMD or KEDMD for the first step to quantify distance between dynamics has two main merits. The first is that instead of applying linear methods, EDMD or KEDMD can capture the nonlinear properties of the dynamics since we consider dynamics in the nonlinearly mapped space of observables. As nonlinearity is one of the key factors of dynamical systems including neural circuits (Freyer et al., 2012), our method may be useful for that purpose. The second is that the matrices used for comparison have clear mathematical meaning as an approximation of the Koopman operator, which is the linear operator in nonlinearly transformed dynamics. This provides better interpretability of what are being compared between dynamics.

Here, note that these Koopman approximant matrices $K_1$ and $K_2$ are of different dimensions. Readers might think that the dimensions should be the same when applying EDMD, since the number of functions in a dictionary should be the same. However, for simpler and more flexible computation, one can reduce the number of functions in a dictionary in a data-driven manner when using either EDMD or KEDMD. For example, one can apply the almost linearly dependent test in EDMD and KEDMD (Engel et al., 2004), where one discards redundant bases in terms of almost linear dependency. This reduction technique is recently applied in the Koopman operator literature (Baddoo et al., 2022). Or one can utilize Nystrom methods for curtailing the row & column numbers of the kernel matrices $G$ and $A$. After these procedures, the dimension on which we project in the functional space ($N$ in the section 2.2) for estimating $K_1$ can be different from that of $K_2$. We put $K_1 \in \mathbb{R}^{m_1 \times m_1}$ and $K_2 \in \mathbb{R}^{m_2 \times m_2}$.

Before calculating the distance between the matrices that are approximation of each dynamics, we have to transform the matrices such that the representation is grounded on orthonormal bases. Eq. (4) implies that $K$ represents transformation of coefficients with respect to the basis $\{\psi_i\}_{i=1}^m$ between adjacent time steps ($m = m_1$ or $m_2$). To compare $K_1$ and $K_2$ in a reasonable manner, we must align the representation such that the basis becomes orthonormal. To get an orthonormal basis $\{u_i\}_{i=1}^m$

of span $(\psi_1, \cdots, \psi_m)$, one can employ QR decomposition in the functional space: $(\psi_1 \cdots \psi_m) = (u_1 \cdots u_m) =: (u_1 \cdots u_m)R$, where the upper triangular matrix $R$ is nonsingular. The vectors $(\psi_1 \cdots \psi_m)$ and $(u_1 \cdots u_m)$ are not members of $\mathbb{R}^m$ but of the $m$-powered product space $\mathcal{F} \times \cdots \times \mathcal{F}$, but the matrix product is calculated according to normal matrix multiplication. For $h = \sum_{i=1}^{m} \tilde{c}_i u_i$ $(\tilde{c}_1, \cdots, \tilde{c}_m \in \mathbb{R})$, its image through $\mathcal{K}'$ is calculated as

$$\mathcal{K}'h = \mathcal{K}'\left((u_1 \cdots u_m)^{\mathsf{t}}(\tilde{c}_1 \cdots \tilde{c}_m)\right) = \mathcal{K}'\left((\psi_1 \cdots \psi_m)R^{-1}{}^{\mathsf{t}}(\tilde{c}_1 \cdots \tilde{c}_m)\right) \qquad (10)$$

$$= (\psi_1 \cdots \psi_m)KR^{-1}{}^{\mathsf{t}}(\tilde{c}_1 \cdots \tilde{c}_m) = (u_1 \cdots u_m)RKR^{-1}{}^{\mathsf{t}}(\tilde{c}_1 \cdots \tilde{c}_m), \qquad (11)$$

implying that the representation with respect to the orthonormal basis $\{u_i\}$ is given as $RKR^{-1} =: \tilde{K}$. In application, $R$ is explicitly calculated as the Cholesky decomposition of $G$, since ${}^{\mathsf{t}}RR = {}^{\mathsf{t}}R(\langle u_i, u_j \rangle)_{ij}R = (\langle \psi_i, \psi_j \rangle)_{ij} = G$. We compute $\tilde{K}$ for each dynamics $\{x_n\}$ and $\{y_n\}$ and denote it by $\tilde{K}_1$ and $\tilde{K}_2$, respectively.

**Modified Procrustes analysis of vector fields (M-PAVF)**     Using the matrices $\tilde{K}_1 \in \mathbb{R}^{m_1 \times m_1}$ and $\tilde{K}_2 \in \mathbb{R}^{m_2 \times m_2}$ in the previous section, we move on to quantifying the metric between the two matrices. Note that these matrices are generally not of the same size: we assume $m_1 \geq m_2$ without loss of generality. A simple and naive extension of PAVF (Eq. (9)) would be

$$d_{\mathrm{P}}(\tilde{K}_1, \tilde{K}_2) := \inf_{U \in \mathcal{O}(m_1)} \left\| \tilde{K}_1 - U \begin{pmatrix} \tilde{K}_2 & 0 \\ 0 & 0 \end{pmatrix} {}^{\mathsf{t}}U \right\|_F, \qquad (12)$$

or, equivalently,

$$d_{\mathrm{P}}(\tilde{K}_1, \tilde{K}_2) := \inf_{U \in \mathrm{St}(m_2, m_1)} \| \tilde{K}_1 - U\tilde{K}_2 {}^{\mathsf{t}}U \|_F, \qquad (13)$$

where $\mathrm{St}(m_2, m_1)$ is a Stiefel manifold of size $m_1 \times m_2$, i.e., $\mathrm{St}(m_2, m_1) = \{U \in \mathbb{R}^{m_1 \times m_2}; {}^{\mathsf{t}}UU = I_{m_2}\}$ and $I_{m_2}$ is the identity matrix of size $m_2$.

Although $d_{\mathrm{P}}$ might look like a felicitous metric between square matrices of different sizes, it comes with a theoretical and a practical shortcomings. The theoretical problem of $d_{\mathrm{P}}$ is that it does not always satisfy the triangle inequality $d_{\mathrm{P}}(A, B) \leq d_{\mathrm{P}}(B, C) + d_{\mathrm{P}}(C, A)$, which disqualifies $d_{\mathrm{P}}$ as a metric (or, more accurately, pseudometric) in a mathematical sense. This can be unfavorable in applying diverse machine learning algorithms after obtaining distances (Williams et al., 2021). The practical demerit is that to obtain $d_{\mathrm{P}}$ one has to resort to optimization algorithms such as neural networks (Ostrow et al., 2023) or iterative optimization over Stiefel manifolds (Edelman et al., 1998; Sato & Iwai, 2013). This can cause a long computational time, especially when the dimensions of the matrices are large. These problems are discussed in the next subsection 3.2 in detail.

Instead, we add a small modification to Eq. (13), where we adjust $\tilde{K}_1$ and $\tilde{K}_2$ with two Stiefel elements:

$$d_{\mathrm{MP}}(\tilde{K}_1, \tilde{K}_2) := \inf_{U, V \in \mathrm{St}(m_2, m_1)} \| \tilde{K}_1 - U\tilde{K}_2 {}^{\mathsf{t}}V \|_F. \qquad (14)$$

We call this metric **Modified Procrustes analysis of vector fields (M-PAVF)**. In fact, this optimization problem has an analytical optimal value, and can be written as

$$d_{\mathrm{MP}}(\tilde{K}_1, \tilde{K}_2) = W_2 \left( \{\sigma_i^{(\tilde{K}_1)}\}_{i=1}^{m_1}, \{\sigma_j^{(\tilde{K}_2)}\}_{j=1}^{m_2} \right), \qquad (15)$$

where $\{\sigma_i^{(\tilde{K}_1)}\}_{i=1}^{m_1}$ and $\{\sigma_j^{(\tilde{K}_2)}\}_{j=1}^{m_2}$ are the sets of singular values of $\tilde{K}_1$ and $\tilde{K}_1$, respectively, and $W_2$ is the $L^2$-Wasserstein distance. Note that when $m_1 \neq m_2$ and for example $m_1 > m_2$, one needs to zero-pad the smaller set of singular values. Using this analytical expression, one can immediately prove that $d_{\mathrm{MP}}$ defines a proper metric on the space of finite dimensional square matrices.

## 3.2   PAVF vs. M-PAVF

In this subsection, we give our experimental results for comparison of the distance defined by PAVF ($d_{\mathrm{P}}$) and M-PAVF ($d_{\mathrm{MP}}$).

**Triangle Inequality**     Here we give an example that $d_{\mathrm{P}}$ cannot satisfy the triangle inequality while $d_{\mathrm{MP}}$ always does. Take $A = \begin{pmatrix} -2 & 0 \\ 1 & 0 \end{pmatrix}$, $B = 2$, $C = -2$. Then computing $d_{\mathrm{P}}$ (Eq. (13))

Table 1: Values of $\rho$ and property of Lorenz system. Properties of systems were based on Sparrow (1982). Description of periodic orbits such as $[1\text{-}2\text{-}2]^2$ represents that the orbit travels around the first equivalent point for one time and the second two times, repeated two times, then comes back to the same orbit. Orbits of these parameters are shown in Appendix Fig. 4.

| $\rho$ | Dynamical property | $\rho$ | Dynamical property |
|---|---|---|---|
| 10 | Line-like trajectory | 100.5 | Periodic $[1\text{-}2\text{-}2]$ |
| 20 | Ring-like trajectory | 100.93 | Intermittent chaos |
| 28 | Chaos | 115 | Chaos |
| 45 | Chaos | 126.5 | Periodic $[1\text{-}1\text{-}2\text{-}2\text{-}1\text{-}2]$ |
| 60 | Chaos | 140 | Chaos |
| 75 | Chaos | 152 | Periodic $[1\text{-}1\text{-}2\text{-}2]$ |
| 99.4 | Chaos | 165 | Periodic $[1\text{-}1\text{-}2\text{-}2]$ |
| 99.537 | Periodic $[1\text{-}2\text{-}2]^8$ | 195 | Chaos |
| 99.6 | Periodic $[1\text{-}2\text{-}2]^4$ | 220 | Periodic $[1\text{-}1]$ |
| 99.7 | Periodic $[1\text{-}2\text{-}2]^2$ | 250 | Periodic $[1\text{-}1]$ |

gives $d_{\mathrm{P}}(A, B) \approx 2.920$, $d_{\mathrm{P}}(B, C) = 4$, $d_{\mathrm{P}}(C, A) \approx 0.7265$, which breaches the triangle inequality. On the other hand, computing our new metric $d_{\mathrm{MP}}$ (Eq. (14)) results in $d_{\mathrm{MP}}(A, B) \approx 0.2361$, $d_{\mathrm{MP}}(B, C) = 0$, $d_{\mathrm{MP}}(C, A) \approx 0.2361$, which preserves the triangle inequality as expected. As satisfying the triangle inequality may be preferred in applying various machine learning techniques in later analyses (Williams et al., 2021), $d_{\mathrm{MP}}$ may be a better choice in quantifying distance between matrices.

**Computational Time**  We next compared computational time needed for $d_{\mathrm{P}}(A, B)$ and $d_{\mathrm{MP}}(A, B)$ through numerical experiment. We randomly sampled 100 pairs of $(A, B) \in \mathbb{R}^{100 \times 100} \times \mathbb{R}^{30 \times 30}$, whose each element follows the Gaussian distribution of mean zero and variance one, and recorded time for computing $d_{\mathrm{P}}(A, B)$ and $d_{\mathrm{MP}}(A, B)$. The result, shown in Fig. 2(a), reveals computing $d_{\mathrm{MP}}$ was on average about 400 times faster than $d_{\mathrm{P}}$. This light computational load can be favorable in examining high dimensional systems, such as neuronal circuits or human brain.

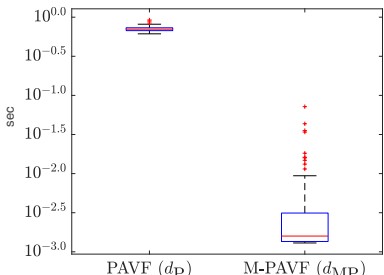

Figure 2: **Computational time of** $d_{\mathrm{P}}$ **and** $d_{\mathrm{MP}}$ **using** 100 **random samples.**

### 3.3 Application to Lorenz Systems

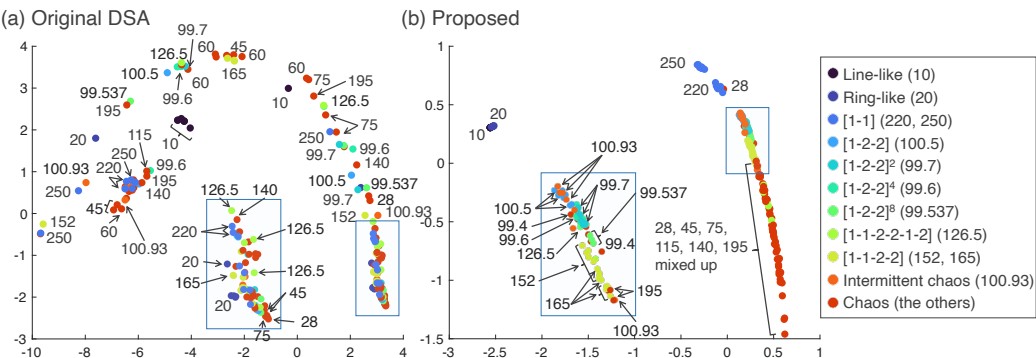

Figure 3: **Two-dimensional projection of distances using original DSA & our method for Lorenz systems.** (a) MDS projection using original DSA. (b) MDS projection using our method. Each color represents the qualitative dynamical property of that parameter.

We next applied our method to simulated dynamics to see if it can capture dynamical property or dynamical similarity. We did this using the Lorenz system (Lorenz, 1963): $\frac{dx}{dt} = \sigma(y - x)$, $\frac{dy}{dt} = x(\rho - z) - y$, $\frac{dz}{dt} = xy - \beta z$. On simulation, we fixed $\sigma = 10$ and $\beta = 8/3$ and changed $\rho$ to obtain various dynamical properties of the system. We chose 20 values of $\rho$ according to the properties of dynamics. The values of $\rho$ and the dynamical properties that $\rho$ give are on Table 1. We performed simulation from the time $t = 0$ to $t = 400$ seconds with a time interval of 0.001 second from a common initial value $(-8, 8, 27)$, and discarded data for the first 200 seconds. Then to collect several data for each parameter, we picked eight periods of the length of five seconds, $[200 + 25k, 205 + 25k]$, $k = 0, \cdots, 7$, for each $\rho$. In this way we obtained 160 time series data as a whole (= 20 values of $\rho \times$ eight time periods), each of which is described as an $\mathbb{R}^{3 \times 5000}$ matrix.

We then applied our method to every pair of these 160 time series data to find dynamical distances. Before application, we normalized the time series by dividing the whole time series by its average norm. For an approximation of the Koopman operators, we applied KEDMD and set the positive definite kernel as the Gaussian kernel of variance 1 ($k(x, y) = \exp\left(-|x - y|^2/2\right)$). We utilized the almost linear dependency test (Engel et al., 2004; Baddoo et al., 2022) to curtail the number of dimensions of Gram matrix $G$. For comparison, we also performed DSA in the same manner, with $h = 100$ points of time delay embedding and the SVD truncation dimension $r = 120$. This gives two 160-by-160 distance matrices, which are shown in Appendix Fig. 5(a) and (d). These distance matrix show distinct tendencies, which represents our method and DSA are extracting different features from dynamics. Two-dimensional mappings of dynamics obtained by employing classical multidimensional scaling (cMDS) are shown in Fig. 3. Each color of points represents each dynamical property described in Table 1 and thus contains one or more than one values of $\rho$. With our method, we can see clusters of points of the same parameters, while DSA fails to. The mapping using our method also shows that points of similar dynamical properties are placed close, as represented in colors. The results imply that our proposed version of DSA may work better than the original version in identifying and distinguishing dynamics of similar dynamical properties.

To check whether KEDMD or M-PAVF was effective, we evaluated distance between dynamics in the following ways: (a) PAVF using HAVOK (i.e., original DSA), (b) M-PAVF using HAVOK, (c) PAVF using KEDMD, and (d) M-PAVF using KEDMD (i.e., proposed method). The results are shown in Appendix Fig. 6. The panels (a) and (b) show that there are no clusters with respect to parameters or dynamic properties, while the panels in (c) and (d) there exist cluster-like structures. These results imply that applying KEDMD was significant in revealing cluster-like structures of dynamics.

## 4 DISCUSSION

In this study, we proposed a new method, a modified version of DSA, which is a data-driven distance quantification between dynamics. Our method quantifies as a distance between dynamics the distance of the Koopman approximant matrices defined in Eq. (14). The optimization problem defined in Eq. (14) boils down to the Wasserstein distance between singular values of the Koopman operator approximants represented with respect to orthonormal coordinates, and thanks to that analytical expression, we can find distance between dynamics at little computational cost.

Although our method reveals clusters when dynamics is periodic as in Fig. 3, it failed in identifying and distinguishing parameters of chaotic dynamics. Although the prediction of chaotic systems is thought to be difficult due to its sensitivity to initial values (Strogatz, 2000), it is desirable if our method could extract chaotic dynamical properties better and identify dynamics of the same parameter. We are currently under investigation of hyperparameters and kernel choices that can distinguish chaotic systems well.

We only applied our method to Lorenz systems in this paper, but our method has broader applicational directions as its target, and we are on our way to pursuing them. This includes classifying neural dynamics between human subjects performing a behavioral task, revealing learning rules between deep neural networks (DNN), or comparing human and DNN neural systems. Through these applications to various neural data, our method might help examining learning rules of the human brain system, not discarding nonlinear properties of dynamics.

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

# A APPENDIX

## A.1 TRAJECTORIES OF LORENZ SYSTEMS

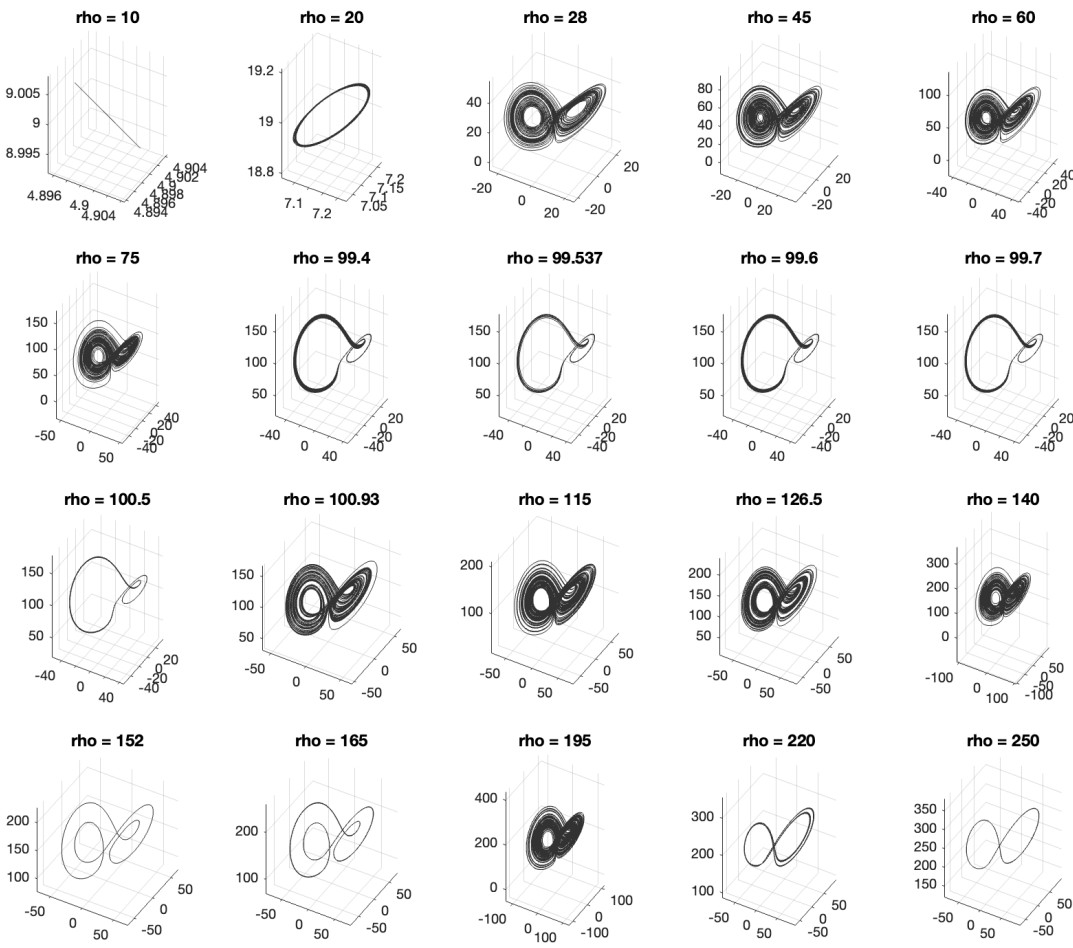

Figure 4: **Trajectories of various values of $\rho$'s.**

### A.2   HAVOK/KEDMD × PAVF/M-PAVF

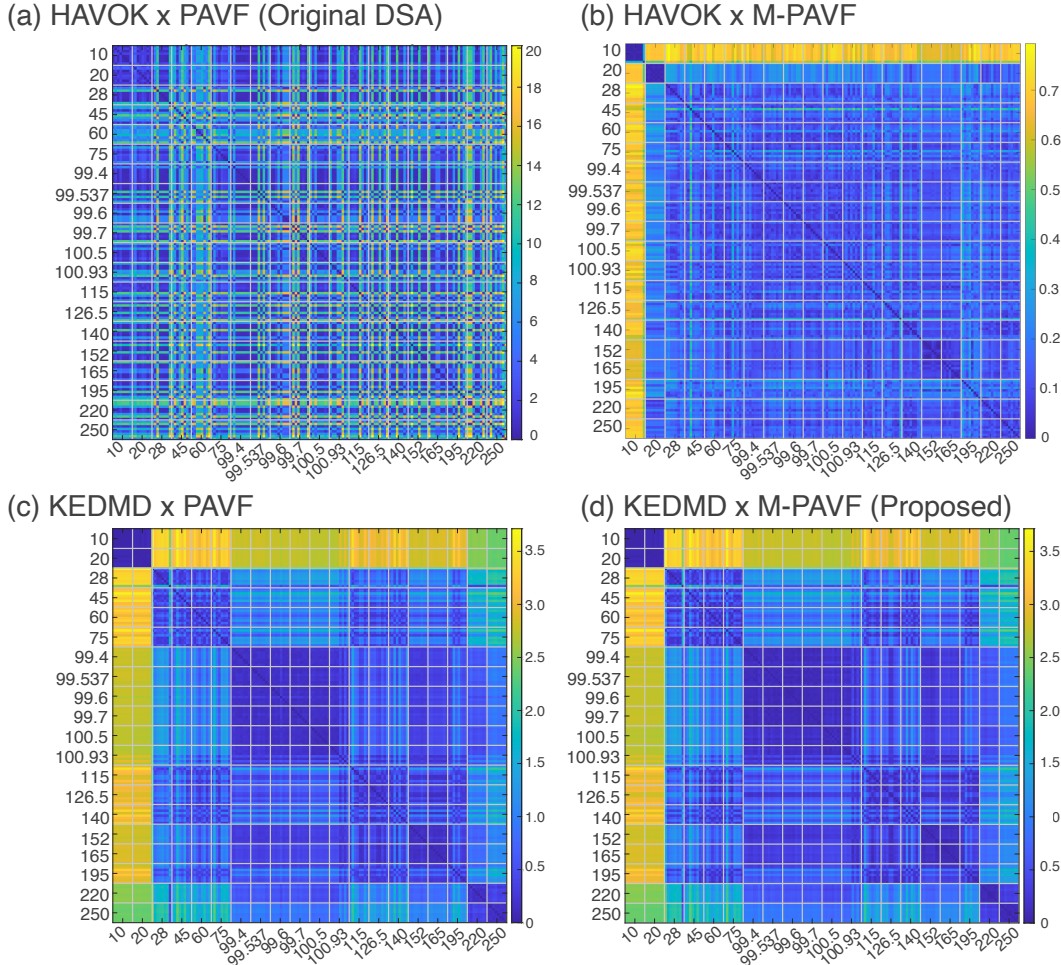

Figure 5: **Distance matrices of four possible methods.** (a) PAVF using HAVOK (i.e., original DSA), (b) M-PAVF using HAVOK, (c) PAVF using KEDMD, (d) M-PAVF using KEDMD (i.e., proposed method). Each small square separated by white borders is an eight-by-eight matrix, whose each element represents the distance between one of eight trajectories simulated with a parameter and one with another.

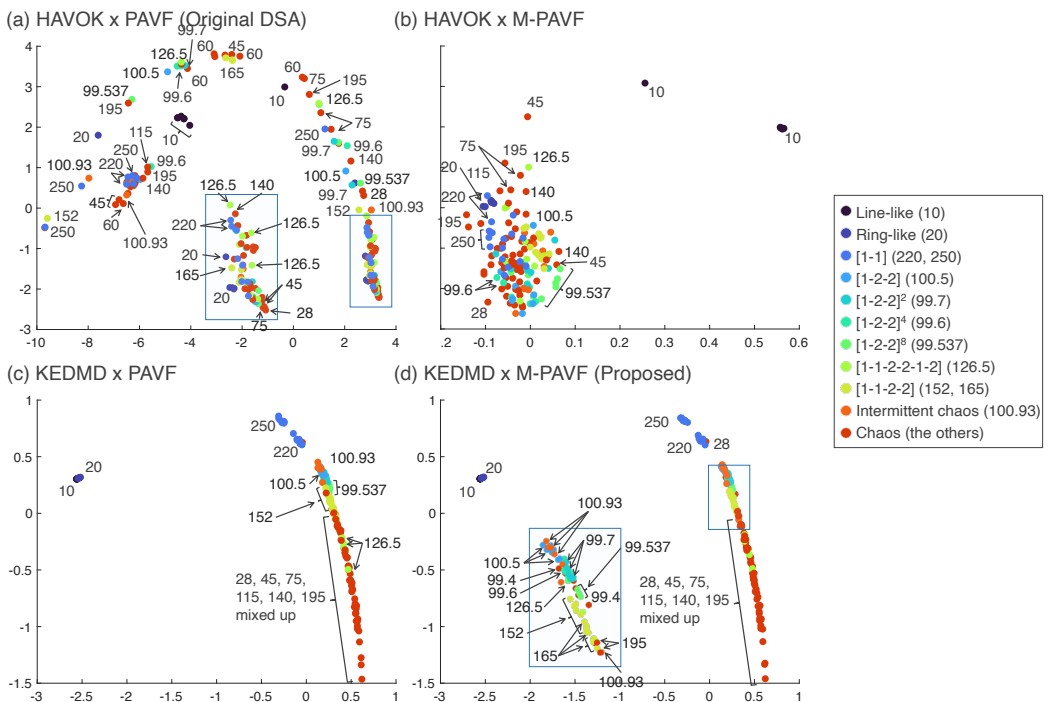

Figure 6: **cMDS results of four possible methods.** (a) PAVF using HAVOK (i.e., original DSA), (b) M-PAVF using HAVOK, (c) PAVF using KEDMD, (d) M-PAVF using KEDMD (i.e., proposed method). Each color represents the qualitative dynamical property of that parameter. (a) and (d) are identical to Fig. 3

