# OpenReview forum: "Koopman Operator Based Dynamical Similarity Analysis for Data-driven Quantification of Distance between Dynamics"
_ICLR.cc/2024/Workshop/Re-Align — ICLR 2024 Workshop Re-Align Poster_

### Official Review · Reviewer_3pmy · 2024-02-21
**Nice work building upon recent results in the field**

**Rating:** 2
**Fit:** 3
**Confidence:** 3

**Workshop Review:**

This paper proposes some tweaks to Dynamical Similarity Analysis (DSA), a recent method introduced by Ostrow et al. (2023; NeurIPS). The bulk of the paper is an explanation of recent developments in Koopman operator theory. The exposition is clear to a general audience; this is an important contribution since the literature being reviewed is recent and I know of no existing authoritative review.

The proposed extensions/reformulations of DSA make sense to me and are generally convincing. I would like the authors to include a derivation of equation 15 (i.e. prove that the modified Procrustes metric has a closed form.

The results / empirical demonstration of this new method need to be fleshed out more, but I think the current submission is sufficient for a workshop.

**Reason For Not Giving Higher Score:**

While the paper is a nice explanation and relevant to this workshop, not enough applications and empirical results are given for me to recommend it for a talk.

**Reason For Not Giving Lower Score:**

Dynamical similarity analysis is an interesting recent development and this paper gives a thoughtful critique of this method. It is clearly relevant to this workshop and clears the bar for quality.

**Reviewer Domain:**

neuroscience

---

### Official Review · Reviewer_VKxd · 2024-02-24
**A modified version of DSA**

**Rating:** 2
**Fit:** 3
**Confidence:** 2

**Workshop Review:**

This paper proposes a modified version of Dynamic Similarity Analysis (DSA) to quantify distance between two dynamical systems. Compared to the original DSA, the proposed method has better interpretability, satisfies the triangle inequality, and has fast computational time.

**Strengths**
- The modifications on the original DSA are novel and well motivated.
- The experiment on Lorenz systems demonstrates that the proposed method captures dynamical properties more effectively than previous methods.
- The method is able to calculate the distance for Koopman approximate matrices of different dimensions.

**Weaknesses**
- While having the distance satisfy the triangle inequality is mathematically appealing, it is not clear what the benefits are in applications.
- While the experiments on Lorenz systems seem promising, the authors did not explain why they evaluate their methods on this specific system. Is it a good proxy to real world systems?

**Reason For Not Giving Higher Score:**

The system used in experiments and the motivation of choosing this particular system could be explained in more detail.

**Reason For Not Giving Lower Score:**

The proposed method is effective and should be interesting to this community.

**Reviewer Domain:**

machine learning

---

### Official Review · Reviewer_KPpZ · 2024-02-26
**Specific methodological improvements for dynamical similarity analysis show promising initial empiric results in interpretability and tractable implementation, but the paper needs clarifying and restructuring**

**Rating:** 2
**Fit:** 3
**Confidence:** 2

**Workshop Review:**

Clarity
* Conceptually clear, and in particular limitations of DSA and how these are addressed by the proposed method are well-explained.
* This structure of DSA problems (1 interpretability, 2 triangle inequality, 3 computational time) is not maintained without. The paper would benefit in terms of clarity and could be made shorter where needed were this the case, including stating in methods which parts of the modified DSA are likely to address which problem. Figure 1 in particular is a great addition in this regard, and the paper should follow that structure.
* Paper is poorly written typographically, linguistically and grammatically: multiple errors of grammar, tense inconsistency, missing commas, missing articles, and other issues limit ease-of-read. I do not list these all as there are numerous, but a careful read-through should be done. Notably, these improve later in the paper (background -- but worse again in section 2.5 -- and results sections).
* I would consider reordering the background section to first discuss DSA, HAVOK, and PAVF as the existing approach, and then the techniques that are employed in the paper -- this would assist with the flow and restating the approach for the reader.

Correctness
* I am not very familiar with the content, but it seems correctly defined mathematically and follows a highly logical flow. E.g. Kernel methods extension is clearly motivated and explained.
* More explanation and intuition of PAVF would help (is it like an alignment transformation then distance?).
* 3.2: should provide a proof that triangle inequality is satisfied rather than example (could be in supplement) -- it is mentioned that m-PAVF is a true metric in section 3.1. I actually think the example can be dropped entirely and the statement made alone.
*3.2: computational time -- what are the costs in terms of accuracy, if any? Given the structure using Cholesky decomposition, you may be able to make theoretical estimates. Whether this is shown empirically or theoretically, I think computational time should be discussed after showing the method works adequately, i.e. after the toy example Lorenz system section 3.3 (and again, to follow the structure of the DSA problem list introduced at the start).
* More discussion of differences in Fig 5 and 6 would be appreciated and interesting; consider restructuring as ablation studies.
* Overall, I think there is too much background and too little discussion of the new method to fully evaluate correctness. It could be interesting to follow the analyses in the Ostrow et al. paper directly.

Novelty
* Methods approach is novel and addresses problems in the existing method literature. Clear acknowledgement of what is new, from where it is adopted, and how it is applied.

Interest
* Relevant new modifications to an existing technique for dynamical systems analysis.

**Reason For Not Giving Higher Score:**

Rating:
* Please see details above.
* Overall, these preliminary results on a new method seem very promising, however the paper was difficult to read and needs careful restructuring and proof-reading, with more time for theoretical and empiric backing showing the utility of the method.
* If these are done, it could still be a good talk paper (I do not know well enough how widely DSA is employed in practice, but the computational time improvements are promising in terms of impact in this domain).

Fit: N/A

**Reason For Not Giving Lower Score:**

Rating:
* Significant new method with initial promising results, that meets criteria in terms of correctness and novelty despite some need for review and improvement.

Fit:
* Clear fit; technical but mathematical details well-explained and of interest.

**Reviewer Domain:**

neuroscience

---

### Decision · Program_Chairs · 2024-03-02

Accept (Poster)